# Effects of the Replacement Length of Concrete with ECC on the Cyclic Behavior of Reinforced Concrete Columns

**DOI:** 10.3390/ma14133542

**Published:** 2021-06-25

**Authors:** Jung-Hwan Hyun, Jin-Wook Bang, Bang-Yeon Lee, Yun-Yong Kim

**Affiliations:** 1Civil Engineering Team, Ssangyong Engineering and Construction Co., Ltd., 299 Olympic-ro, Songpa-gu, Seoul 05510, Korea; hyun686@ssyenc.com; 2R&D Center, Tongyang Construction Materials Co., Ltd., 2822 Gimpo-daero, Gimpo 10024, Korea; jinwook.bang@tongyanginc.co.kr; 3School of Architecture, Chonnam National University, 77 Yongbong-ro, Buk-gu, Gwangju 61186, Korea; bylee@jnu.ac.kr; 4Department of Civil Engineering, Chungnam National University, 99 Daehak-ro, Yuseong-gu, Daejeon 34134, Korea

**Keywords:** column, cyclic behavior, ductility, energy dissipation, ECC

## Abstract

This paper presents an experimental investigation on the effects of the replacement length of concrete with engineered cementitious composites (ECC) on the cyclic behavior of a reinforced concrete (RC) column. A conventional RC column specimen and two RC composite columns designed with ECC were fabricated. To investigate the cyclic behavior of each specimen, a series of cyclic loading tests was performed under a reversed cyclic loading condition with a constant axial load. Test results showed that ECC columns exhibited higher cyclic behavior in terms of load carrying capacity, ductility, and energy dissipation capacity compared to the RC column. It was also found that when applying ECC to the column specimen with a length of 3.6*d* or more, the energy dissipation capacity was greatly increased.

## 1. Introduction

As the economic growth and development of construction technology has been rapidly expanding recently, there has been an increasing demand for huge structures, such as high-rise buildings, long span bridges, and huge underground structures. Although concrete is one of the most widely used construction materials in the construction industry due to its low cost and durability, it has drawbacks, such as low tensile strength, which is approximately 10% of its compressive strength, and inherent brittle failure under tension load. An engineered cementitious composite (ECC) has been developed to overcome the inherent disadvantages of normal concrete [1,2]. ECC shows high tensile ductility based on multiple micro-cracks by incorporating synthetic fibers below 2 vol.%; the micromechanics and steady-state cracking theory are adopted for its material selection and mixture design [3,4,5,6]. Many researchers have tried to utilize this material in structural members and retrofitting [7,8,9,10,11,12,13,14,15].

Fischer and Li [16] investigated the cyclic and deformation behavior of steel reinforced ECC flexural members. Test results showed that the combination of a ductile cementitious matrix and steel reinforcement results in improved energy dissipation capacity, reduction of steel reinforcement requirements, and damage-tolerant inelastic deformation behavior compared with a normal reinforced concrete (RC) member. Fischer and Li [17] also investigated deformation behavior of fiber-reinforced polymer-reinforced ECC flexural members under reversed cyclic loading conditions. The interaction of linear elastic FRP reinforcement and ECC matrix with ductile behavior in tension results in nonlinear elastic flexural response characteristics with stable hysteretic behavior, low residual deflection, and ultimately, gradual compression failure. Salahuddin et al. [18] evaluated the feasibility of using ECC as a means to enhance the performance of beam-column connections. It was found that the use of ECC in the plastic zone of the connection and partial replacement of transverse reinforcement significantly enhanced the joint shear resistance, energy absorption capacity, and cracking response. Cho et al. [19] investigated the cyclic responses of RC composite columns strengthened in the plastic hinge region locally by high performance fiber reinforced cementitious composites (HPFRC), which were designed to improve the seismic performance. Test results showed that the application of HPFRC to RC columns not only improved cyclic lateral load and deformation capacities, but also minimized cracks in the critical region of the RC columns. Although the feasibility of the structural application of ECC to flexural members and columns was demonstrated, the study on the effects of concrete’s replacement ratio with ECC on the cyclic behavior of an RC composite column is fairly limited.

Therefore, the purpose of this study is to experimentally investigate the effect of the replacement ratio of concrete with ECC at the plastic hinge region of columns on the cyclic behavior of RC composite columns.

## 2. Experimental Program

### 2.1. Materials

ECC and normal concrete were used as a matrix material in this study. Type I ordinary Portland cement (Tongyang cement industrial Co., Ltd., Seoul, Korea) with a density of 3.15 g/cm^3^ and a Blain fineness of 3570 cm^2^/g was used. The density and Blaine fineness of fly ash (F1tech co., Ltd., Boryeong, Korea) were 2.15 g/cm^3^ and 3590 cm^2^/g, respectively. The density and Blaine fineness of blast furnace slag (F1tech co., Ltd., Boryeong, Korea) used as a binder were 2.89 g/cm^3^ and 4230 cm^2^/g. PVA fiber (Kuraray co., Ltd., Tokyo, Japan) was used as a reinforcing fiber and its physical properties are listed in Table 1. The amount of surface coating with oil was 1.2%, which was intended to reduce excessive chemical bonding between the fiber and the matrix [20]. Polycarboxylate-based superplasticizer (Kerneos Inc., Paris, France) and hydroxypropyl methylcellulose (Avebe co., Ltd., Veendam, The Netherlands) were used to control the rheological properties of the matrix and achieve homogenous fiber dispersion. A defoamer blended with hydrocarbons and polyglycols was also used to prevent unintentional excessive pores and the formation of large pores.

The mixture proportions of ECC and concrete used in this study are listed in Table 2 and Table 3. ECC was designed to have a high tensile strain capacity of over 3%. Figure 1 shows the specimen geometry of ECC for the uniaxial tension test, which was performed in accordance with JSCE recommendations [21]. In order to induce 3-D random distribution of fibers in specimens, the thickness of specimen was set to 30 mm. Figure 2 shows the tensile stress-strain curves of ECC at an age of 28 days. As shown in Figure 2, all specimens showed strain-hardening behavior and a high tensile strain capacity of 3.2%. In all specimens, the crack width was controlled less than 100 μm and multiple micro cracks were observed. The compressive strength of ECC was 21 MPa at an age of 28 days. Normal concrete with the compressive strength identical to that of ECC was also designed and prepared for manufacturing the composite column. Two types of deformed reinforcing bars with D13 and D10 were used as longitudinal and transvers reinforcing bars. Both reinforcing bars have a yield strength of 300 MPa.

### 2.2. Design and Manufacturing Process of Column Specimens

Figure 3 shows the dimension and details of the specimens designed in this study. One specimen of the conventional RC column and two specimens of the RC composite columns (ECC_3_._6*d*_ to ECC_5_._4*d*_) were manufactured. The replacement length (*L_R_*) of concrete with ECC for each specimen is listed in Table 4. All specimens are composed of three parts: the head for applying the load, the main column, and the base. Eight D13 reinforcing bars were used as longitudinal bars; and D10 reinforcing bars with a spacing of 100 mm in the main column were arranged as transverse shear reinforcing bars. The main column was embedded 40 mm into the base for the fixed condition at the end, and the base was fixed to the strong floor. Figure 4 shows the manufacturing process of the specimens. First, formworks were fabricated and reinforcement rebars were installed. After that, the strain gauges were attached to the rebars for measuring the strain. Then, ECC and concrete were mixed and poured into the mold. All column specimens were cured for 28 days under dry curing conditions at an average temperature of 21.6 °C before laboratory tests were carried out.

### 2.3. Experimental Procedure

Figure 5 shows the experimental set-up for investigating the cyclic behavior of the composite columns. The axial force of 20 kN corresponding to 10% of the axial load-bearing capacity of the column was applied to all specimens using a hydraulic actuator (SAMYEON Co., Ltd., Daegu, Korea) with a capacity of 300 kN, and the lateral cyclic load was applied using a hydraulic actuator with a capacity of 500 kN. A steel plate was attached to the head of the column to prevent failure of the head due to stress concentration. The lateral cyclic load was applied to the top of the column by displacement control. A displacement transducer (TML Co., Ltd., Tokyo, Japan) was installed at the top of the column to measure and control the displacement of the column. Figure 6 shows the cyclic loading history of columns. Two cycles at each step according to the drift ratio were programmed.

## 3. Results and Discussion

### 3.1. Crack and Failure Patterns

Figure 7 shows the crack patterns of all specimens after the cyclic loading tests. In the RC specimen, the initial flexural crack occurred in columns within 1.0*d* from the column-base joint at the lateral displacement of 4.3 mm. Additional flexural cracks appeared near the column-base joint, as the cyclic load increased. The width of the initial flexural crack increased and flexural cracks occurred near the column-base joint when the lateral displacement was larger than 26.1 mm. As the lateral displacement level increased, several macro cracks occurred up to the top of the column by enlarging micro cracks. At the lateral displacement of 34.8 mm, a maximum lateral loading state of the RC specimen was observed and the crack width at the column-base joint increased. As the cyclic loading level increased, shear cracks occurred in the cover concrete of the column near the zone of the column-base joint. A rapid reduction of the load bearing capacity of the joint also occurred. Until the lateral displacement reached 52.2 mm, the spalling of the cover concrete and buckling of the longitudinal reinforcing bars were observed in the plastic hinge region within 0.84*d* (210 mm) from the column-base joint. The plastic hinge length was calculated using the formula suggested in the previous study [22].

For the ECC_3_._6*d*_ specimen, an initial micro flexural crack was observed near the bottom column at the lateral displacement level of 4.3 mm. As the lateral displacement increased, the number of micro cracks gradually increased and reached a maximum load at the lateral displacement of 40.6 mm after flexural cracking in the column-base joint. In comparison to the RC specimen, no macro cracks were observed at the column until the maximum load was reached. At the lateral displacement of 61.2 mm, the ECC_3_._6*d*_ specimen finally failed, with multiple micro cracks near the height of 3.6*d* from the column-base joint. Unlike the RC specimen, spalling of the cover concrete and the buckling of longitudinal reinforcing bars near the plastic hinge zone was not observed. From the cracking pattern observation, it was found that the replacement of ECC effectively prevented the plastic hinge damage generated in the RC specimen under cyclic loads. The ECC_5_._4*d*_ specimens showed similar fracture behavior to the ECC_3_._6*d*_. As the ECC replacement length increased, the number of micro cracks increased and the crack spacing decreased. Both ECC_3_._6*d*_ and ECC_5_._4*d*_ specimens exhibited effective prevention of shear cracks, spalling of the cover concrete, and the buckling of the longitudinal reinforcing bars, compared to the RC specimen.

### 3.2. Hysteretic Behavior of Column Specimens

Figure 8 shows the hysteresis curves of columns; Table 5 lists the performance of columns under the positive force condition. The initial cracking lateral load of the RC specimen was 14.2 kN, and rebar yielding and maximum lateral load were measured to be 23.6 kN and 32.2 kN, respectively. After the maximum load, the load-bearing capacity of the column-base joint decreased suddenly and the test was terminated at the lateral displacement of 52.2 mm.

On the other hand, the initial cracking load values of the ECC_3_._6*d*_ and ECC_5_._4*d*_ specimens were increased by 1.1 kN and 1.7 kN, respectively, compared to the RC specimen. This indicates that the increase of initial cracking load was insignificant with an increase in the *L_R_*. As the lateral displacement increased, the lateral force of ECC specimens increased. The longitudinal reinforcement of the column-base joint yielded at the range of load lever from 25.7 to 28.7 kN, which were higher than that of the RC specimen by 8.9% and 21.6%, respectively. As the *L_R_* increases, the lateral displacement measured at the top of the column also increased. The maximum load values of ECC_3_._6*d*_ and ECC_5_._4*d*_ specimens were increased by 9.3% and 21.4% higher than that of the RC specimen. The lateral displacement at maximum load state also increased by 23.0% to 31.8% compared to the RC specimen. This indicates increased ductility of the columns through the use of ECC. The ultimate displacement ($\delta_{u}$) of the ECC series specimens tended to increase along with the *L_R_* of concrete with ECC. Therefore, it was found that the seismic performance of the column-base joint increased as the *L_R_* of concrete with ECC increased.

The ductility of the specimens can be evaluated by calculating the ductility ratio, which is the ratio of ultimate displacement ($\delta_{u}$) to yield displacement ($\delta_{y}$). The ductility ratio of the RC, ECC_3_._6*d*_, and ECC_5_._4*d*_ specimens were 2.88, 3.44 and 3.40, respectively. Compared to the RC specimen, the ECC series showed an improved ductility ratio by minimizing the spalling of the cover concrete and the buckling of the longitudinal reinforcement. Although the ultimate displacement of the ECC specimens increased with an increase in the *L_R_*, all ECC specimens exhibited a similar ductility ratio. This means that the effect of *L_R_* on the ductility ratio is not significant. The increase of the ductility ratio of ECC specimens is mainly due to tightly controlled bending and shear cracks in the plastic hinge zone through the strain hardening behavior and multiple micro cracks of ECC. This resulted in an increase in the displacement of ECC specimens at the yielding.

### 3.3. Energy Dissipation Capacity

The energy dissipation capacity of structures under cyclic loading conditions is an important factor for evaluating the seismic behavior of structural members. The energy dissipation of all specimens is shown in Figure 9. Before the longitudinal reinforcement yielded, all the specimens were measured at a similar level of energy dissipation. However, energy dissipation increased rapidly after a yield of longitudinal reinforcement and plastic deformation of the cover concrete near the column-base joint. ECC series specimens showed increased maximum load and displacement while minimizing the joint damage. The cumulative energy dissipation of the ECC specimens, i.e., ECC_3_._6*d*_ and ECC_5_._4*d*_ were 41.7% and 101.3% higher than that of the RC specimen, respectively.

Figure 10 shows the ductility ratio and cumulative dissipated energy for each specimen. The ductility ratio and cumulative energy dissipation capacity of the ECC series specimens are higher than those of the RC specimen. Although the ductility ratio tended to be relatively insignificant due to the increased yielding displacement of the ECC series specimens, the energy dissipation capacity, which is the total area of the load and displacement relation curve, greatly increased with a length of 3.6*d* or more. From the test results, it was found that the energy dissipation capacity and the strengthening effect were enhanced as *L_R_* increased. In the construction field, however, economic efficiency and required strengthening level should be considered when we select a suitable *L_R_*.

## 4. Conclusions

In this study, the effects of the replacement length of concrete with ECC were experimentally evaluated under reversed cyclic loading conditions. An RC specimen and two types of ECC specimens with different replacement lengths of concrete with ECC were fabricated and a series of cyclic load tests was carried out. From the test results, the following conclusions were drawn:In the RC specimen, as the lateral displacement increased after the maximum load, the load decreased drastically. Local damage was also observed, such as spalling of the cover concrete near the plastic hinge on the length of 1.0*d* from the column-base joint, and buckling of the longitudinal reinforcing bars. In comparison to the RC specimen, ECC specimens exhibited effective prevention of shear cracks and spalling of the cover concrete, and the buckling of longitudinal reinforcement was not serious. Excellent tensile properties of the ECC, such as strain hardening behavior and multiple micro cracks, led to a minimization of spalling of the cover concrete by controlling flexural and shear cracks in the plastic hinge zone of the column.In the case of the ECC specimens, the ductility ratio of the RC, ECC_3_._6*d*_, and ECC_5_._4*d*_ specimens were 2.88, 3.44 and 3.40, respectively. Although the ultimate displacement of the ECC specimens increased with an increase in the *L*_R_, all ECC specimens exhibited a similar ductility ratio. This means that the effect of *L*_R_ on the ductility ratio is not significant. The increase of the ductility ratio of ECC specimens is mainly due to tightly controlled bending and shear cracks in the plastic hinge zone. This resulted in an increase of displacement of ECC specimens at the yielding.The energy dissipation capacity was improved up to 101.3% higher than that of the RC specimen. Based the results of this experiment, it is recommended that the replacement length of concrete with ECC from the column-base joint is considered to be more than 3.6*d* at the plastic hinge region of an ordinary RC column.

## Figures and Tables

**Figure 1 materials-14-03542-f001:**
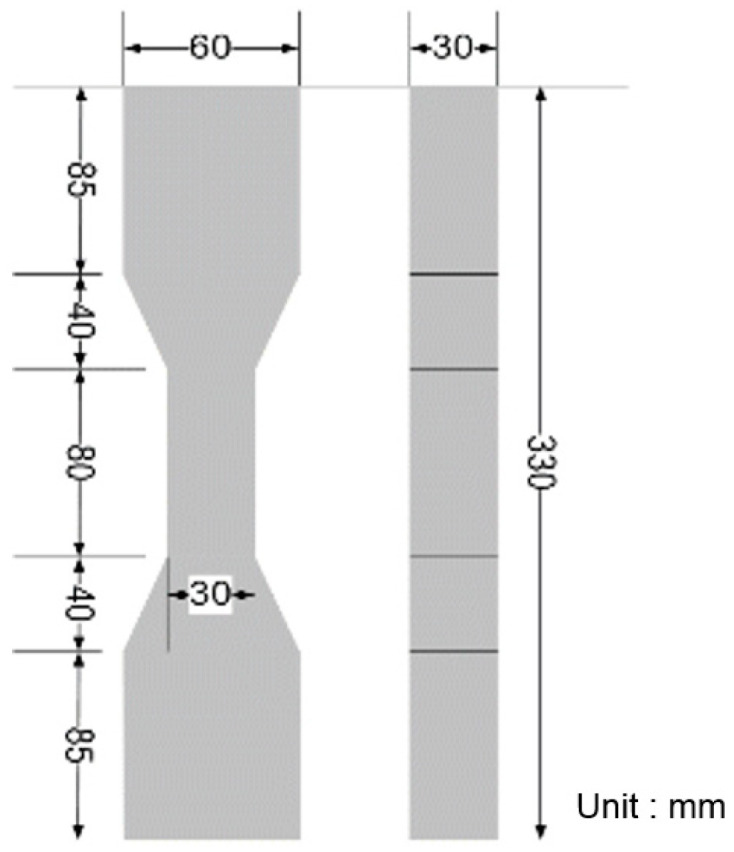
Dimension of ECC specimen for uniaxial tension test.

**Figure 2 materials-14-03542-f002:**
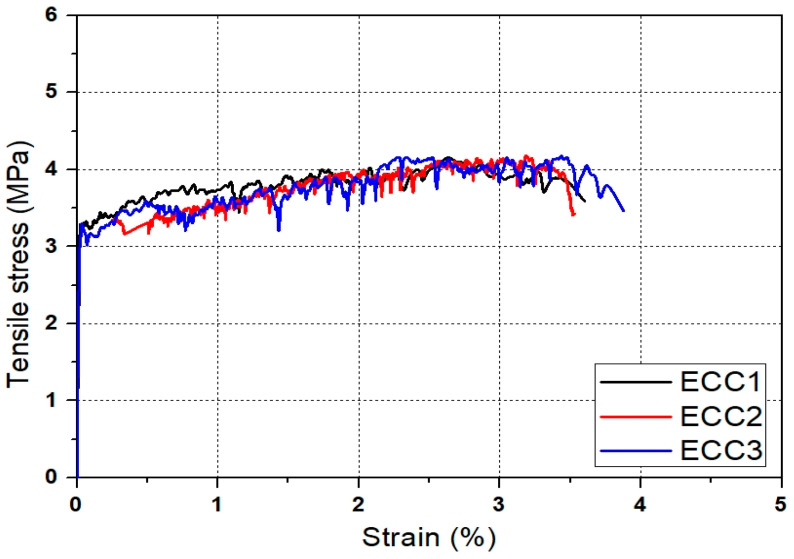
Tensile stress-strain curves of ECC.

**Figure 3 materials-14-03542-f003:**
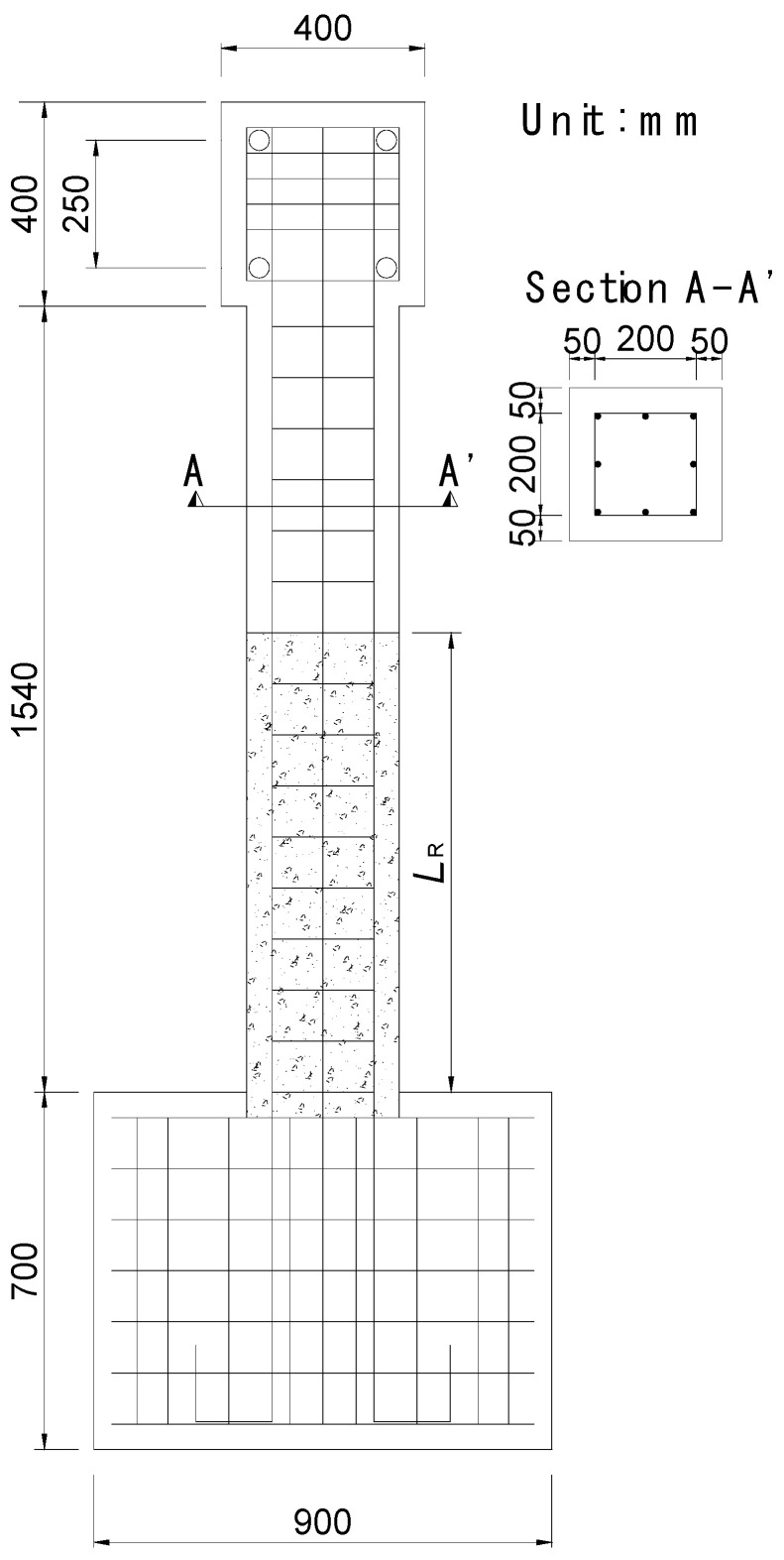
Geometry and dimension details of the column specimen.

**Figure 4 materials-14-03542-f004:**
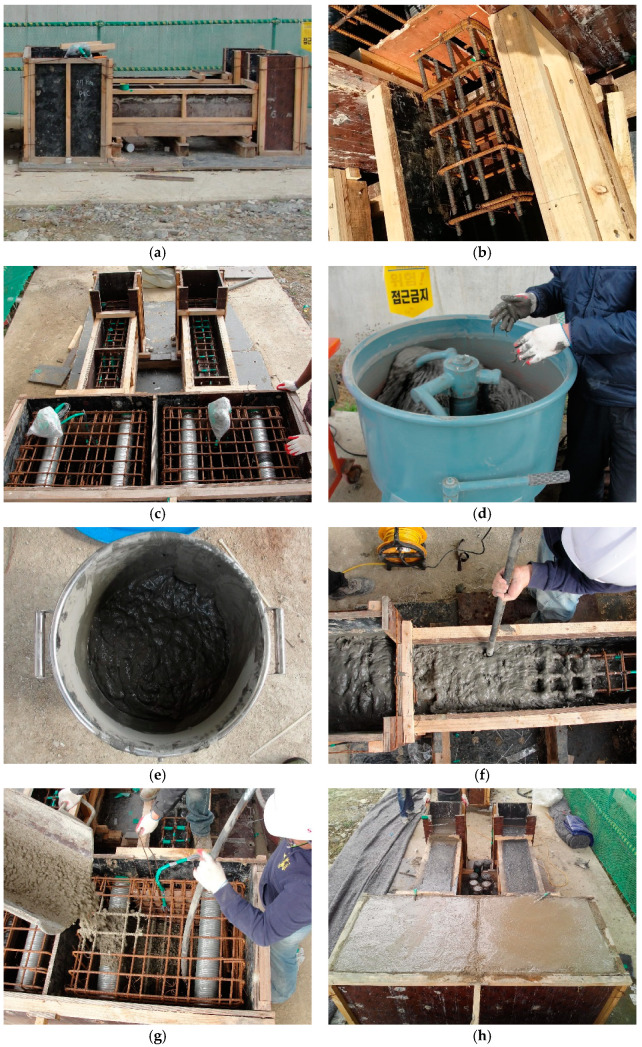
Manufacturing process of column specimens (**a**) Formwork (**b**) Rebar installation (**c**) Strain gauges installation (**d**) ECC mix (**e**) Mixed ECC (**f**) Placement of ECC (**g**) Placement of concrete (**h**) Finish (**i**) Curing of specimen (**j**) Manufactured specimen.

**Figure 5 materials-14-03542-f005:**
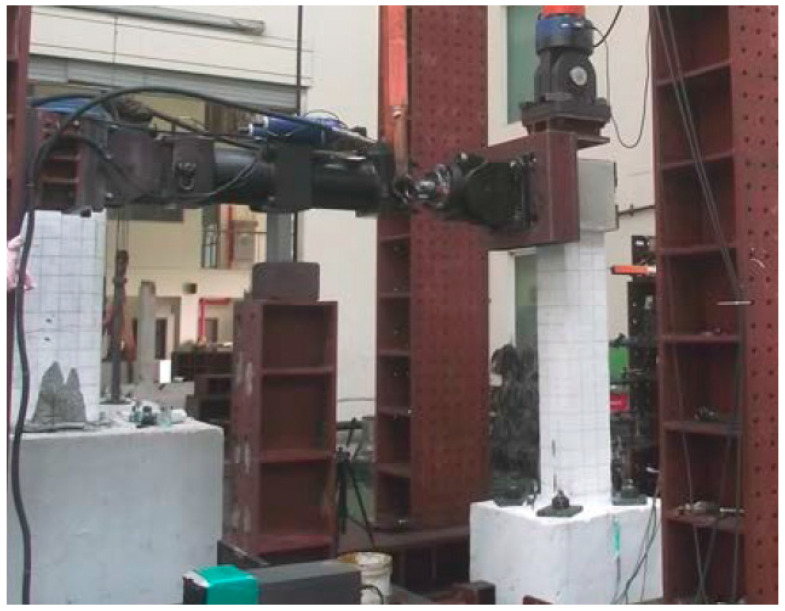
Setup for cyclic loading test.

**Figure 6 materials-14-03542-f006:**
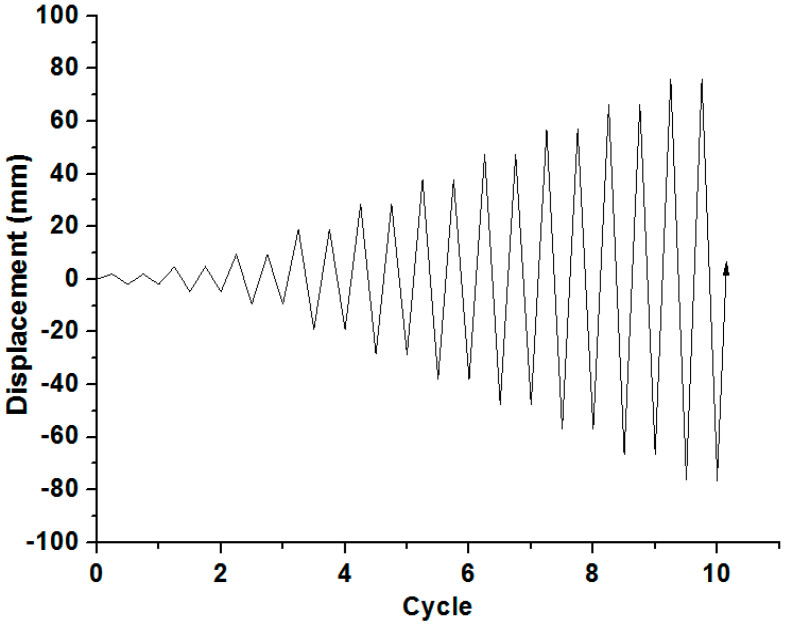
Cyclic loading history by displacement control.

**Figure 7 materials-14-03542-f007:**
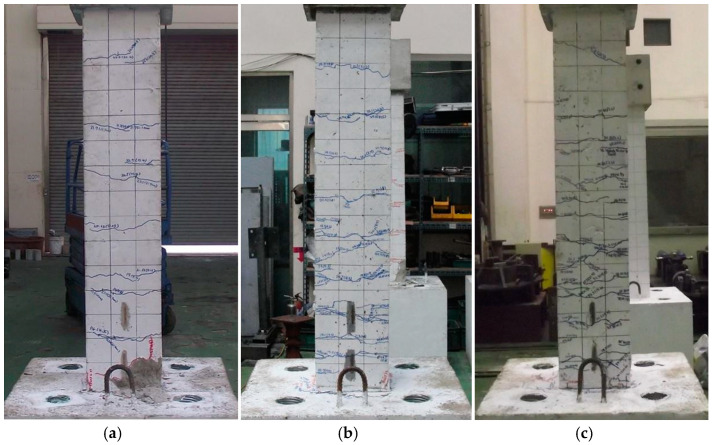
Final crack and failure patterns of specimens (**a**) RC; (**b**) ECC_3_._6*d*_; (**c**) ECC_5_._4*d*_.

**Figure 8 materials-14-03542-f008:**
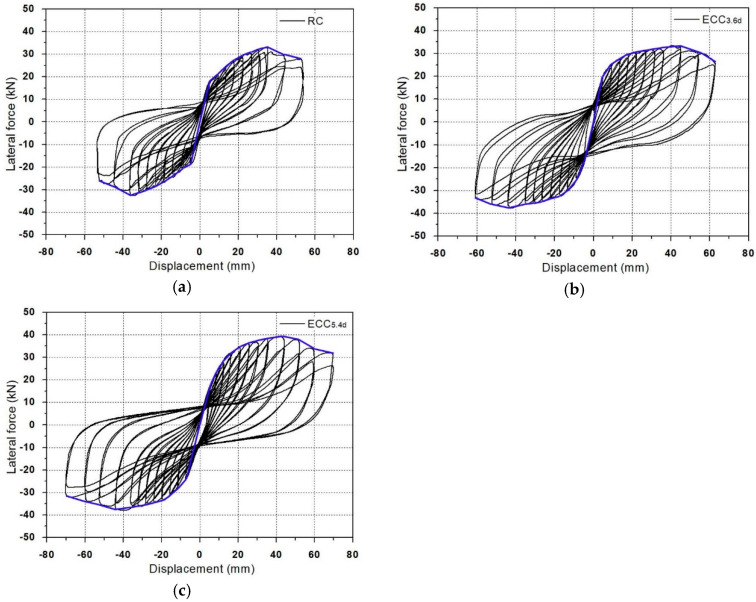
Relationship curves of lateral load–displacement hysteretic behavior (**a**) RC (**b**) ECC_3_._6*d*_ (**c**) ECC_5_._4*d*_.

**Figure 9 materials-14-03542-f009:**
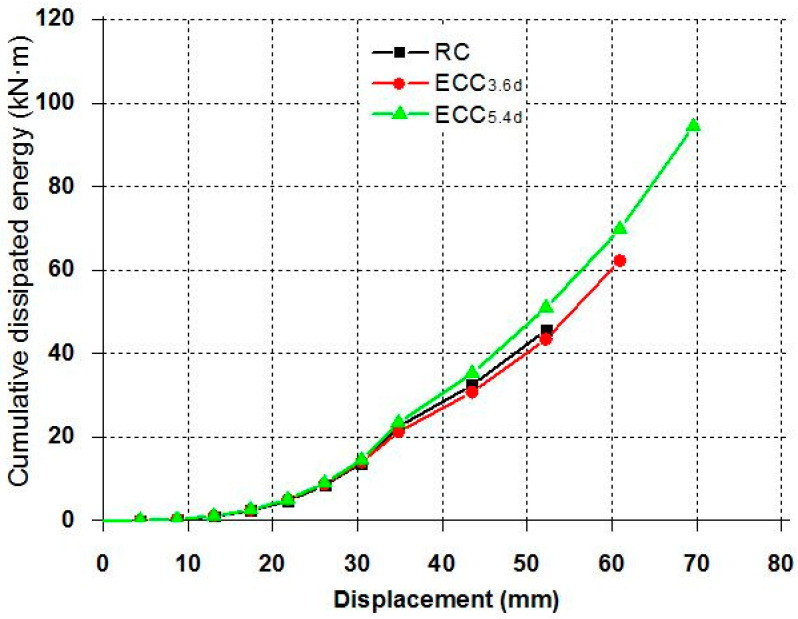
Cumulative dissipated energy of each specimen.

**Figure 10 materials-14-03542-f010:**
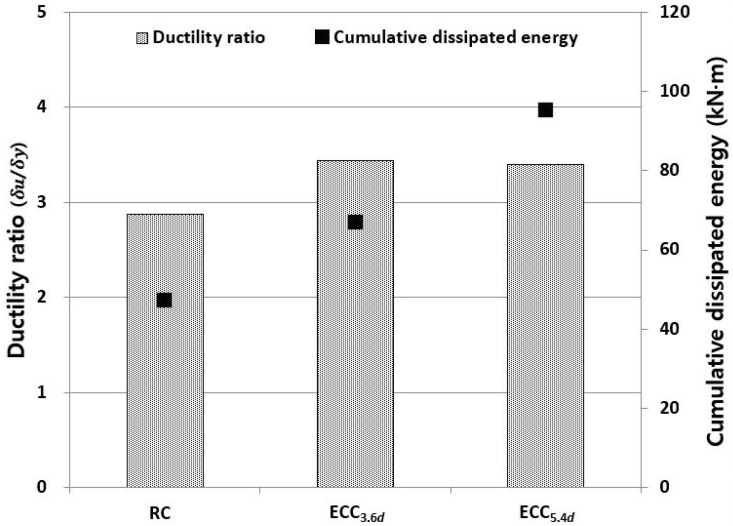
Ductility ratio and cumulative dissipated energy for each specimen.

**Table 1 materials-14-03542-t001:** Properties of PVA fiber.

Density (g/mm^3^)	Length (mm)	Diameter (μm)	Tensile Strength (MPa)	Young’s Modulus (GPa)	Breaking Elongation (%)
1.3	12	39	1600	40	6.5 (dry)

**Table 2 materials-14-03542-t002:** Mixture proportion of ECC.

Unit: kg/m^3^						
W	OPC	FA	SP	HPMC	Deformer	PVA Fiber (Vol.%)
362	411	1152	0.494	0.165	0.041	2.0

Note: W: Water, OPC: Ordinary Portland Cement, SP: super plasticizer, HPMC: hydroxypropyl methyl cellulose, PVA: polyvinyl alcohol.

**Table 3 materials-14-03542-t003:** Mixture proportion of concrete.

W/B	S/a	Unit: kg/m^3^						Ad.
		W	OPC	BFS	FA	S	G	
53.8	48.9	167	216	47	47	892	931	2.2

Note: BFS: Blast furnace slag, S: Sand, G: Gravel (maximum size of 15 mm), Ad.: Admixture.

**Table 4 materials-14-03542-t004:** Details of column specimens.

Specimen	ECC Replacement Length from Column-Base Joint (mm)	Longitudinal Reinforcing Bar	Transverse Reinforcing Bar
RC	-	8-D13	D10@100
ECC_3_._6*d*_	900	8-D13	D10@100
ECC_5_._4*d*_	1350	8-D13	D10@100

**Table 5 materials-14-03542-t005:** Result of repeated loading test under the positive force condition.

**Specimen**	**Initial** **Cracking** **State**		Rebar Yielding State		Maximum Loading State		Ultimate Loading State		Ductility Ratio $(\delta_{u}$ $/\delta_{y})$
	$\delta_{i}$ **(mm)**	$P_{i}$ **(kN)**	*δ_y_* (mm)	*P_y_* (KN)	*δ_m_* (mm)	*P_m_* (KN)	*δ_u_* (mm)	*P_u_* (KN)	
RC	4.3	14.2	17.2	23.6	33.0	32.2	49.6	26.5	2.88 (100%)
ECC_3_._6*d*_	4.3	15.3	18.6	25.7	40.6	35.2	63.9	30.5	3.44 (119.4%)
ECC_5_._4*d*_	4.4	15.9	20.5	28.7	43.5	39.1	69.3	32.2	3.40 (118.1%)

## Data Availability

Not applicable.

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
