# Peer review of "Effects of the Replacement Length of Concrete with ECC on the Cyclic Behavior of Reinforced Concrete Columns"

_materials, 2021, doi:10.3390/ma14133542_

Round 1
Reviewer 1 Report
Dear authors:
It contains some weaknesses and mistakes which must be eliminated by introducing minor revision according to the suggestions given below.
1 ECC has many other merits. Please add more recent references about ECC.
Ref.:
Review of Cementitious Composites Containing Polyethylene Fibers as Repairing Materials. Polymers 2020, 12, 2624.
2. L66. Please clarify the type and properties of cement and fly ash.
3. L71. Please clarify the resource and properties of SP, HPMC and defoamer.
4. L81. Please provide the dry breaking elongation of PVA fibers.
5. L92. How to connect the ECC column and the concrete base? Please provide more details.
6. L106. What is the meaning of ECC replacement length? Please mark it in Figure 3.
7. L116. Why do you choose 500 kN as the lateral cyclic load? Please explain.
Author Response
We appreciate your sincere and prudent comments. In the manuscript, the revised parts have been marked in underlined typeface. We hope that the revised version of this paper will satisfy your comments.
Please check the attached file for responses to the reviewer's comments.

Reviewer 2 Report
This paper conducted an experimental investigation on the effects of the replacement length of concrete with ECC on the cyclic behavior of RC columns. The results are useful and can extend the database of the structural behavior of the ECC component. The manuscript can be considered for publication after addressing the following comments and suggestions.
1# For the ordinary concrete used, the mix proportion (including the aggregate size) and the value of compressive strength should be provided.
2# It is nice to use the 30-mm thick tensile specimen for ECC applied in structural element, because the fiber distribution in such a specimen can be considered as a 3-D random distribution. This point is suggested to be pointed out in the manuscript.
3# Again, for the direct tension of ECC, the thickness of ECC was set as 30 mm, which is different from the conventional materials’ level test (i.e., 13-mm thick in JSCE). It is suggested to provide more information about the tensile properties of ECC (e.g., crack width and crack spacing).
4# In the mix proportion of ECC, the water-to-binder ratio was 362/(411+1152) = 0.23, which was a comparatively low value. However, the compressive strength of ECC was only 21 MPa. It may not be rational for PVA-ECC. Please check this value and give some explanation.
5# The average environmental temperature during the 28-day curing should be provided.
6# The skeleton curves for cyclic loading specimens need to be marked in Fig. 8.
7# For clarity, the replacement length of concrete with ECC is suggested to be marked in Fig. 3.
8# In the introduction, some recent advances of ECC applied in the structures/components subject to cyclic loading are suggested to be considered (e.g., <doi.org/10.1016/j.compstruct.2019.04.061> and <doi.org/10.1016/j.engstruct.2019.109576>).
Author Response

(The authors gave the same response as above.)

Reviewer 3 Report
The paper deals with the application of the ECC to some part of the RC columns. The concrete was replaced with ECC on the length where the plastic hinge appears during cyclic loading. The test results revealed its effectiveness.
I found the paper valuable. All the issues raised in the paper are clearly presented. Only one information is missing:
- Page 6, line 113 - “The axial force corresponding to 10% of the axial load-bearing capacity of the column was applied..” - Please explain how was the load-bearing capacity of the column determined. Please show the values of the load-bearing capacity of column. Was it assigned experimentally?
Please read the paper carefully. There can be found mistakes in the text, which have to be corrected e.g.
- page 8, line 182 – “On the other hand, the initial cracking strength values of the…”- it should be “load” instead of “strength”.
Author Response

(The authors gave the same response as above.)

Reviewer 4 Report
Very good paper with non-traditional combination reinforced concrete and ECC. I only suggest to describe more the method of putting ECC into formwork and stabilisation of ECC in required place. But this is detail.
Author Response

(The authors gave the same response as above.)
